# Point-estimating observer models for latent cause detection

**Jennifer Laura Lee** *, **Wei Ji Ma** *

Center for Neural Science, New York University, New York City, New York, United States of Amercia

* jll616@nyu.edu (JLL); wm44@nyu.edu (WJM)

## Abstract

The spatial distribution of visual items allows us to infer the presence of latent causes in the world. For instance, a spatial cluster of ants allows us to infer the presence of a common food source. However, optimal inference requires the integration of a computationally intractable number of world states in real world situations. For example, optimal inference about whether a common cause exists based on $N$ spatially distributed visual items requires marginalizing over both the location of the latent cause and $2^N$ possible affiliation patterns (where each item may be affiliated or non-affiliated with the latent cause). How might the brain approximate this inference? We show that subject behaviour deviates qualitatively from Bayes-optimal, in particular showing an unexpected positive effect of $N$ (the number of visual items) on the false-alarm rate. We propose several "point-estimating" observer models that fit subject behaviour better than the Bayesian model. They each avoid a costly computational marginalization over at least one of the variables of the generative model by "committing" to a point estimate of at least one of the two generative model variables. These findings suggest that the brain may implement partially committal variants of Bayesian models when detecting latent causes based on complex real world data.

**Funding:** W.J.M. received funding from the National Institutes of Health (National Eye Institute) Grant No: R01EY020958-09. The funders had no role in study design, data collection and analysis, decision to publish, or preparation of the manuscript.

## Author summary

Perceptual systems are designed to make sense of fragmented sensory data by inferring common, latent causes. Seeing a cluster of insects might allow us to infer the presence of a common food source, whereas the same number of insects scattered over a larger area of land might not evoke the same suspicions. The ability to reliably make this inference based on statistical information about the environment is surprisingly non-trivial: making the best possible inference requires making full use of the probabilistic information provided by the sensory data, which would require considering a combinatorially explosive number of hypothetical world states. In this paper, we test human subjects on their ability to perform a causal detection task: subjects are asked to judge whether an underlying cause of clustering is present or absent, based on the spatial distribution of those items. We show that subjects do not reason optimally on this task, and that particular computational short cuts ("committing" to certain world states over others, rather than

**Competing interests:** The authors have declared that no competing interests exist.

representing them all) might underlie perceptual decision-making in these causal detection schemes.

## Introduction

Many forms of perception or cognition require the inference of high-level categorical variables from a multitude of stimuli. For example, the spatial distribution of visual items allows the perceptual decision-making system to infer the presence or absence of latent causes in the world (a high-level categorical variable). The Bayesian framework for perceptual decision-making takes a "generative models" approach [1], positing perception as inference over a latent state of the world based on noisy data. A generative model specifies how a stimulus may be generated from the presence or absence of combinations of latent causes (or objects) in a scene. The Bayes-optimal observer knows this generative model and uses it to perform inference based on observed sensory data. The Bayesian approach is successful at capturing human decision-making data for many cases of perceptual multisensory cue integration (e.g. [2, 3]) and sensorimotor learning (e.g. [4]), and has also been successful in providing a computational account of various perceptual grouping phenomena [5].

The generative model approach has been useful in tasks that require the grouping or organization of ambiguous image fragments into latent causes. The problem of perceptual grouping can be conceptualized as an instance of the broader problem of detecting a latent cause (e.g., a partially occluded object), or delimiting its effects, based on directly observable stimuli (e.g., image elements). Bayesian hierarchical grouping [6] is one such example of a framework for perceptual grouping based on generative mixture models. Image elements are assumed to be generated by a mixture of distinct objects, with each object generating image elements according to certain generative assumptions, offering an explanation for dot clustering, contour integration, and part decomposition in visual perception. Similarly, in the contour integration literature, researchers have re-interpreted the "association field" (how strongly the visual system associates two line segments with a particular configuration of positions and orientations as belonging to one contour) [7] as the conditional link probability between two oriented line segments based on a generative model [8]. Contour integration– the ability to detect curvilinearly aligned edge configurations despite randomly oriented distractors– was found to be optimal in humans [9]. These findings suggests that the generative model approach may be a useful starting point for constructing observer models that describe how the brain performs these tasks.

Similar processes may underlie our ability to make high-level inferences about hidden states of the world based on graphically abstracted visual data. For example, in the "London bomb problem," an individual looks at a map of bombings to determine whether they are all indiscriminate or whether some bombs are in fact aimed at a common target. In this problem, the ideal observer must consider all possible target locations, and the possibility that each bomb was generated by a targeted vs. random process. The normative Bayesian framework proposed by Griffiths and Tenenbaum [10] provides a unifying account of our sense of coincidence in a variety of contexts— including, for instance, coincidences over space (e.g., many bombs landing in a cluster on the map), time (e.g., many birthdays falling close together in date), and category (e.g., landing many heads in a row in a series of coin flips)– providing an account of how we may decide certain states are mere coincidences while others lead us to infer a common cause. Empirically, they show that a Bayesian model roughly tracks trends in human ratings of coincidence on a variety of causal inference tasks.

                              

However, in each of these cases of perceptual grouping or causal inference, the Bayesian framework provides an approximate "as if" description of human behaviour, and falls short of making any commitments about the mental representations and algorithms actually carried out by the brain during these judgements. The success of Bayes-optimal models does not necessarily entail that humans perform full Bayesian probabilistic computations at the algorithmic or mechanistic level [11, 12].

Moreover, if one were to interpret the computational Bayesian model for spatial coincidence detection as a representation-level model of the inner workings of the brain, the number of computations required to solve a relatively simple spatial coincidence detection task would quickly exceed a number which the visual system might plausibly implement. This is especially problematic for natural visual scenes with a large number of elementary visual components. How might the brain then be implementing something that approximates the Bayesian computation in the case of perceptual grouping and spatial causal inference tasks?

In a Bayesian model, one of the most computationally taxing steps involves "marginalizing" an often high-dimensional probability distribution over a large number of generative model variables (the joint distribution) in order to arrive at the probability distribution for the specific task-relevant variable of interest (the marginal distribution). This step is often intractable in natural settings with many objects and visual features [13]. For example, in the London bombing problem described above, for a map with $N$ visual items, the Bayesian calculation amounts to considering $2^N$ distinct combinations of "targeted" vs. "random" items.

In a decision-making task, these "nuisance" generative model variables are variables of the generative model that do not bear directly on the decision of the observer, but that must be accounted for (as in "marginalized out") in order to arrive at the variable of interest in Bayesian inference. For example, in the London bombing problem, the variable of interest is the binary top-level category (whether or not the bombings are targeted), while an example of a nuisance parameter is the location of the target, if it exists. Taking into account possible identities of the nuisance parameter (e.g. hypothetical target location) is necessary on the Bayesian model to arrive at an optimal estimate about the variable of interest (e.g. presence or absence of a target). The generative model employed by an ideal Bayesian observer is often assumed to correspond to the true underlying process that generates the observations, meaning those nuisance variables and variables of interest are those determined by the true underlying process itself. Previous studies in perceptual decision-making [14, 15] suggest that subjects might use simplified point-estimates of certain nuisance variables instead of marginalizing over their full probability distributions, resulting in particular patterns of suboptimal behaviour (but see [16]). One way to potentially evade the combinatorial explosion of unique representations entailed by the Bayesian computation during spatial coincidence detection is to represent only some task parameters as full probability distributions, while committing to others as single-point estimates. This family of approximately Bayesian models may be considered "partially committal", in that they commit to at least one of the nuisance variables of the generative model as a point estimate, while still framing the problem as one involving inference about generative model variables (rather than taking on an entirely heuristic strategy that requires no knowledge of the generative model at all).

In the current study, we employ a spatial coincidence task inspired by the "London bombing" problem to test whether spatial causal inference is optimal or may be better accounted for by "point-estimating" observer models. Our version of the task uses the spatial distribution of pigeons in a park, affected by a pigeon feeder whose location is not directly observable. Pigeons cluster around the pigeon feeder if she is present. The subject's goal is to infer the presence or absence of the feeder. The generative model of the task entails two abstract parameters: 1) the location of the causal object (feeder) and 2) which of all observations "are affiliated with" (i.e.,

"result from") the causal object. We ask whether these two parameters are represented by the brain at all, and if so, whether they are represented in full probabilistic form, or as collapsed point estimates.

We then test three hypotheses about probabilistic representation in the brain during a spatial coincidence detection task. On the Strong Bayesian representation hypothesis, the brain represents all of the abstract parameters of the generative model, including their full probability distributions. On the Non-Probabilistic representation hypothesis, the abstract variables of the generative model are not mentally represented at all: instead, subjects assess spatial coincidences using some heuristic metric like the mean distance between points. Lastly, the Point Estimate representation hypothesis holds that the variables entailed by the generative model are indeed represented by the brain, but that not all such parameters can be represented as full probability distributions: at least some are represented as single-point estimates.

## Task

10 subjects were given a cluster detection task (Fig 1), in which they were asked whether a set of dots was drawn from a random uniform distribution, or from a mixture of a uniform distribution and a Gaussian. In task context, dots denoted the location of pigeons in a park. A "causal object" was introduced as an invisible "pigeon feeder" whose location was not directly observable. Subjects were instructed as follows: "On days when the pigeon feeder is present, pigeons tend to cluster around her location. But even when the feeder is present, there's only a 50% chance that a given pigeon will be affiliated with her." Pigeons which were not affiliated with the feeder were drawn randomly. The location of the feeder herself was drawn from a Gaussian distribution centered at the center of the screen. Subjects indicated whether a feeder

**Fig 1. Task design and example stimuli.** Example of "feeder present" (top) and "feeder absent" stimuli (bottom) for $N = 10$. The stimulus is shown for 400 ms, after which subjects are prompted to respond about the presence or absence of the feeder, and then asked to rate their confidence level from 1 (least confident) to 4 (most confident).

 

was present by pressing a button. Their decision was based on the spatial distribution of pigeons on the screen. Subjects completed 20 practice trials with full feedback ("correct"/ "incorrect," including an image showing the true underlying partition of pigeons and actual location of the feeder if there was one), followed by the main task, comprised of 2000 trials with partial feedback ("correct"/ "incorrect").

## The generative model

We first describe the generative model of the task. Each trial is experimentally defined as either a "Feeder present" ($C = 1$) or "Feeder absent" ($C = 0$) trial. These trials occur in equal proportions: $p(C = 0) = p(C = 1) = 0.5$. If $C = 0$, each observation (pigeon location) is drawn from a uniform distribution on a disc of radius $R$, centered at $(0, 0)$. If $C = 1$, the pigeon feeder's location $\mu$ is drawn from a circular Gaussian centered at $(0, 0)$ with standard deviation $\sigma_s$, i.e., $p(\mu|C = 1) = \mathcal{N}(\mu; (0,0), \sigma_s^2 I)$, where $I$ is the $2 \times 2$ identity matrix. On these "feeder present" trials, for each of $N$ pigeons, there is a $p_{\text{aff}}$ chance that the pigeon is affiliated with the pigeon feeder (in the experiment, $p_{\text{aff}} = 0.5$). If a pigeon is not affiliated ($z_i = 0$), its location is drawn from a uniform distribution on a disc of radius $R$ centered at the center of the screen, just as in the $C = 0$ case. If the pigeon is affiliated with the feeder ($z_i = 1$), then the pigeon's location is drawn from a smaller Gaussian distribution centered at the location of the pigeon feeder ($\mu$) with standard deviation $\sigma$, such that $p(x_i|z_i = 1, \mu, C = 1) = \mathcal{N}(x_i; \mu, \sigma^2 I)$. As a a result, the distribution of a pigeon location when the feeder is present is a mixture of a Gaussian distribution centered at $\mu$ and a uniform distribution:

$$p(x_i|\mu, C = 1) = p_{\text{aff}}\mathcal{N}(x_i; \mu, \sigma^2 I) + \frac{1 - p_{\text{aff}}}{\pi R^2} \text{ for } ||x_i|| < R.$$

Through our choice of $\sigma_s = 2$ and $\sigma = 2$ (where the park had a circular radius of 10 units), it was exceedingly rare that $||x|| > R$, but when this happened, a new stimulus was drawn.

## Observer models

Despite its apparent simplicity, the pigeon task poses a strong computational challenge to any inference system. This is because of the problem of nuisance parameters, variables that an optimal observer would marginalize over to compute a posterior distribution over the variable of interest. In our task, the generative model entails two nuisance parameters: $\mu$, the location of the bird feeder (if present), and the "partition variable" $\mathbf{z}$, a binary vector that denotes which of the $N$ pigeons are affiliated or unaffiliated with the feeder. The ideal Bayesian observer would represent both $\mu$ and $\mathbf{z}$ as probability distributions. Here, we test variants of the Bayesian observer model, fully heuristic models, and models that commit to (instead of marginalizing over) at least one of these two nuisance parameters.

Given $N$ observations $\mathbf{x} = \{x_1, \ldots, x_N\}$, the observer is tasked to infer category $C$. The log posterior ratio is

$$d = \log\frac{p(C = 1|\mathbf{x})}{p(C = 0|\mathbf{x})} = \log\frac{p(C = 1)}{p(C = 0)} + \log\frac{p(\mathbf{x}|C = 1)}{p(\mathbf{x}|C = 0)} \tag{1}$$

Since $p(C = 0) = p(C = 1) = 0.5$, we have

$$d = \log\frac{L_1}{L_0}, \tag{2}$$

where we have introduced $L_0 \equiv p(\mathbf{x}|C = 0)$ and $L_1 \equiv p(\mathbf{x}|C = 1)$.

 

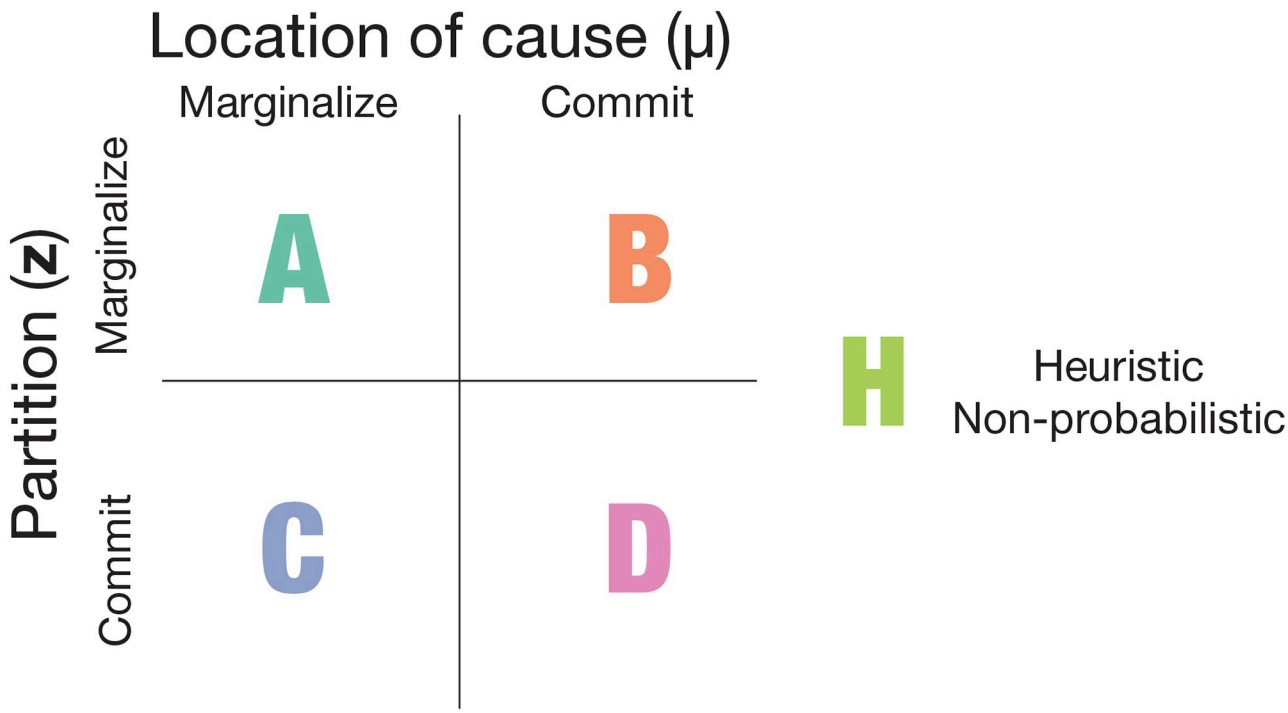

**Fig 2. Taxonomy of model families.** Family H models do not posit the representation of either location or partition.

We test five families of observer models, each with a different assumption about what it is that cognitive systems represent during the task (Fig 2). In Family A (the Strong Bayesian model), the brain represents full probability distributions over both variables of the generative model. $L_1$ is calculated through a double marginalization: by integrating over all possible feeder locations $\mu$ and by summing over all possible $\mathbf{z}$:

$$\text{Family A}: \quad L_1 = \sum_{\mathbf{z}} \int L(\mathbf{z}, \mu; \mathbf{x}) p(\mathbf{z}) p(\mu) d\mu$$

Here, $L(\mathbf{z}, \mu; \mathbf{x}) \equiv p(\mathbf{x}|\mathbf{z}, \mu)$ represents the likelihood of a particular feeder location and partition given a set of pigeon observations. The prior probabilities of each feeder location and partition are independent of one another and multiply this likelihood.

In Family B, the brain commits to and represents only a single feeder location ($\hat{\mu}$), but represents (and sums over) the full probability distribution of partitions ($\mathbf{z}$):

$$\text{Family B}: \quad \tilde{L}_1 = \sum_{\mathbf{z}} L(\mathbf{z}, \hat{\mu}; \mathbf{x}) p(\mathbf{z})$$

For instance, in one model, the brain simply chooses the center of mass of all of the pigeons as its committed $\mu$. For a given pigeon observation, since no marginalization of $\mu$ is needed, and summing over different partitions involves consideration of its "affiliated" and "unaffiliated" likelihoods, $\tilde{L}_1$ can be re-expressed as a product over observed pigeons, where $p(x_i|\hat{\mu})$ can be broken down into a given pigeon's "affiliated" and "unaffiliated" terms: $\tilde{L}_1 = \prod_{i=1}^{N} p(x_i|\hat{\mu}, C = 1)$, where each factor is given by Eq (1).

Conversely, in Family C, the brain commits to and represents only a single partition $\hat{\mathbf{z}}$. That is, it only represents one of all possible combinations of affiliated pigeons, while still

representing the feeder location $\mu$ as a probability distribution over all possible locations:

$$\text{Family C}: \quad \tilde{L}_1 = \int L(\hat{\mathbf{z}}, \mu; \mathbf{x}) p(\mu) d\mu$$

The particular set of pigeons to represent as "affiliated", $\hat{\mathbf{z}}$, is determined by the specifics of each of the models within Family C. For example, in model C8, the chosen $\hat{\mathbf{z}}$ is the partition that maximizes the marginal likelihood, so that $\tilde{L}_1 = \max_{\mathbf{z}} L(\mathbf{z}; \mathbf{x})$. In model C9, the observer commits to $\hat{\mathbf{z}}$ through "agglomerative clustering"– an algorithm that begins by calculating the centroid of the pigeons as a reference point, then iteratively considers increasingly distant pigeons for inclusion into a "feeder affiliated" cluster.

In Family D, the brain represents both feeder location and partition as point estimates rather than probability distributions. The general form of models in this family is:

$$\text{Family D}: \quad \tilde{L}_1 = L(\hat{\mathbf{z}}, \hat{\mu}; \mathbf{x}).$$

For instance, one model commits to both feeder location and partition by maximizing their joint posterior: $\tilde{L}_1 = \max_{\mathbf{z}, \mu} L(\mathbf{z}, \mu; \mathbf{x}) p(\mu)$.

For all of Families A through D, once the log posterior ratio $d$ is calculated, observer responses are modelled allowing for decision noise (Gaussian noise on the log posterior ratio with variance $\sigma_d$), and a lapse rate $\lambda$, which represents the probability of randomly choosing either category with probability 0.5. The ideal Bayesian observer would respond $C = 1$ when $d > k$ where $k = 0$. However, we also allow for the possibility that observers have unequal utilities, costs, or incorrect prior assumptions when reporting $C = 0$ and $C = 1$, by fitting the decision criterion $k$ as a free parameter over all trials or as four separate parameters for each of $N = 6, 9, 12, 15$.

Putting everything together, the probability of reporting $C = 1$ becomes

$$p(\hat{C} = 1|\mathbf{x}) = 0.5\lambda + (1 - \lambda)\Phi\left(\log\frac{\tilde{L}_1}{L_0}; k, \sigma_d^2\right), \tag{3}$$

where $\tilde{L}_1 = L_1$ for Family A and $\Phi$ denotes the cumulative normal distribution.

Lastly, in Family H, the brain does not represent feeder location or partition at all. Instead, observers solve the task by representing some other abstract variable outside of the generative model, like "pigeon density." For instance, the observer might simply represent the density of the points on screen and respond "feeder present" if that number exceeds some threshold. Each heuristic model has the form

$$p(\hat{C} = 1|\mathbf{x}) = \Phi(f(\mathbf{x}); k_n, \sigma_d^2),$$

where the function $f$ represents the model's heuristic.

Family A represents the Strong Bayesian representation hypothesis, Families B, C, and D the Point-Estimate representation hypothesis, and family H the non-probabilistic representation hypothesis. See S1 Fig for the full list of observer models and their parameters by family, and see Methods for a more detailed description of each model.

## Results

Previous studies suggest that observers may combine their observations of ambiguous visual elements with knowledge about a generative model in order to perform inferences about latent causes. The aim of the present study is to investigate how observers represent and compute over the intermediate "nuisance" variables entailed by these generative models, especially

when those computations become intractable. To achieve this, we designed a task paradigm in which observers are asked to detect the presence of a latent cause (a "pigeon feeder") based on a large number of image fragments (the distribution of "pigeons" in a park). We then fit four different families of models, each of which represents the generative model's nuisance variables as either a point-estimate or full probability distribution.

On "feeder absent" trials, the pigeons are drawn from a uniform distribution over the park. On "feeder present" trials, the pigeons are drawn from a mixture of a uniform and a Gaussian distribution, where the Gaussian is centred at the location of the hidden pigeon feeder. When the feeder is present, every pigeon has a 50% chance of affiliation with it (i.e. a 50% chance of having been drawn from the Gaussians).

We first characterize behavioural responses as a function of basic stimulus properties like "pigeon density." As one might expect, subjects are more likely to respond "feeder present" the more densely packed the pigeons are on screen. Intuitively again, when the distance between the two closest pigeons is very small (i.e., when the two closest pigeons are very close together), subjects are also more likely to respond "feeder present." (Fig 3) Nevertheless, subjects are surprisingly adept at telling apart trials when the feeder truly is or isn't present even when statistics like the "mean distance" between points are controlled for across trials. This fact is evidenced by the vertical offset of the red and blue curves in Fig 3. Therefore, from the outset, we see that if a summary statistic like density or minimum distance is represented, they cannot possibly be the only thing that subjects represent in order to make accurate judgements about spatial coincidence in this task.

We next examine the trend in responses, broken down by "feeder present" and "feeder absent" trials, as the number of pigeons ($N$) varies. One unexpected finding is that the proportion of "feeder present" responses increases as a simple function of the number of pigeons ($N$) (Fig 4), even for "feeder absent" trials. In other words, as the number of uniformly drawn dots increases, subjects are more likely to sound a false alarm and report "Feeder present". Examining model predictions, we see that this unexpected qualitative trend is at odds with the Strong Bayesian hypothesis (Fig 4)– intuitively, if we were able to reason optimally, observing more dots drawn from a uniform distribution should make us more certain that the distribution is in fact uniform. Instead, we see that the number of "feeder present" responses increases with increasing $N$. Henceforth, we refer to the slope of this curve as the "effect of $N$ on false alarms."

We next ask whether the strong Bayesian model can be rescued by fitting a different prior over category (or decision criterion) for each $N$. We tested this Bayes variant (see Model A2,

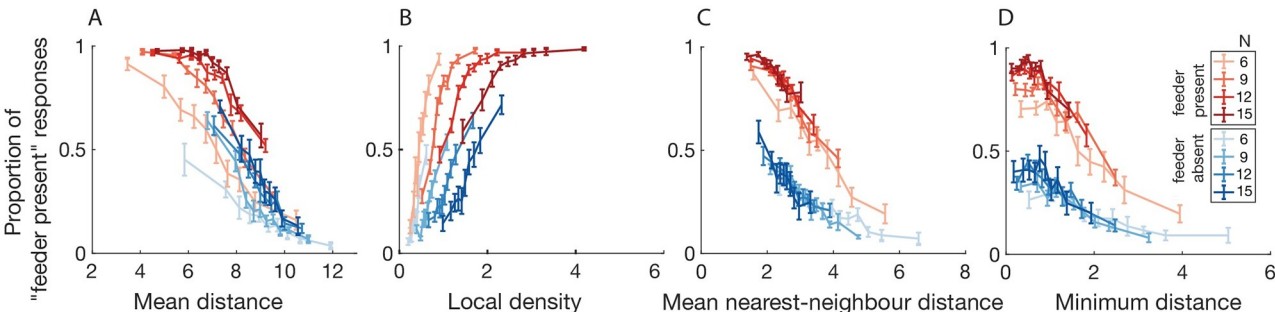

**Fig 3. Proportion of "feeder present" responses as a function of each distance-based heuristic.** "Feeder absent" trials in blue, "feeder present" trials in red, with colour saturation indicating the number of pigeons on a given trial ($N$ = 6, 9, 12, 15). For each trial, the following quantities were computed: the mean pairwise distance of all pigeons (A), the maximum local density, where local density is computed by convolving a gaussian of $\sigma$ = 1.4cm over the circular arena (B), the nearest neighbour distance, calculated as the mean distance between all pigeons and their nearest neighbour (C), and the distance between the two nearest pigeons (D). Points are binned in equal quantiles along the abscissa quantity such that an equal number of trials are represented in each point. For stimulus histogram distributions along each abscissa quantity, see S2 Fig.

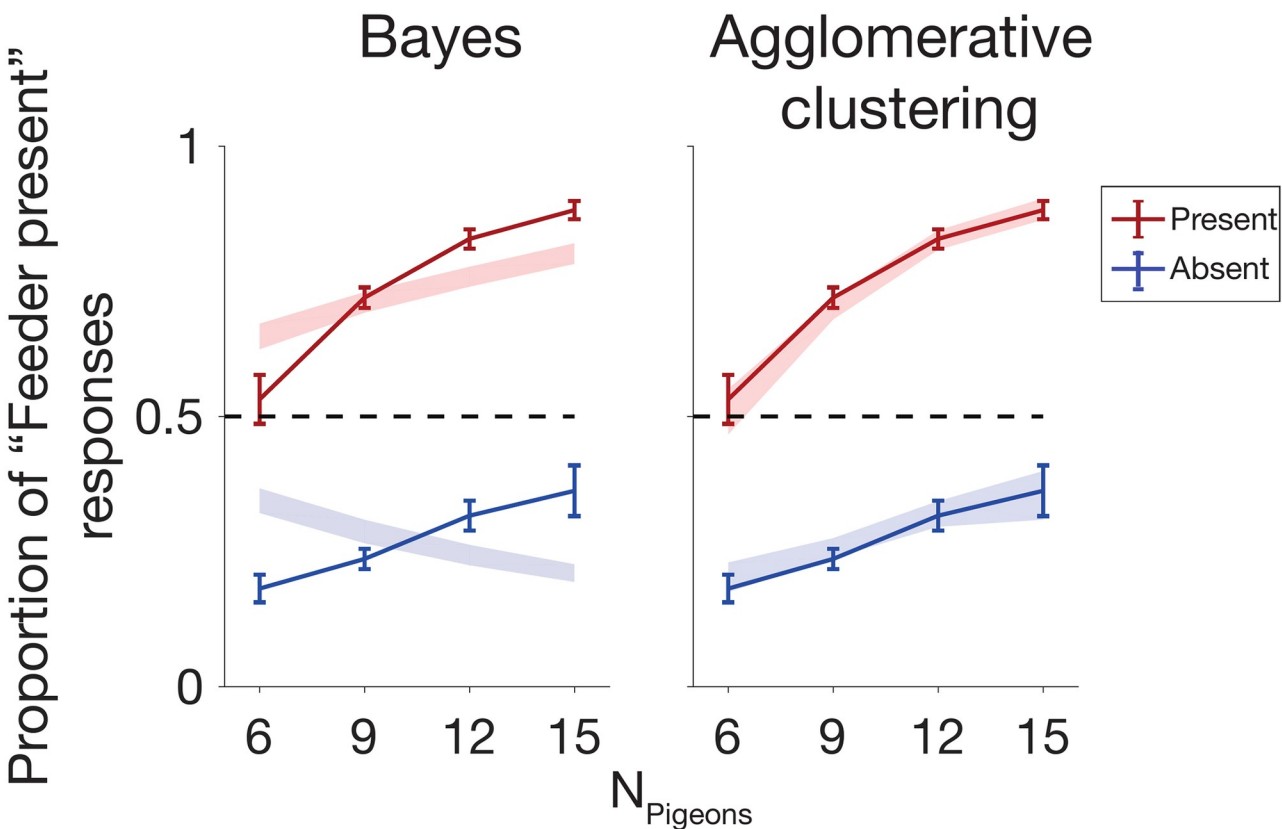

**Fig 4. Model fits of proportion of "feeder present" responses as a function of number of pigeons (N) denoted by shaded area; subject data denoted by solid lines.** The Bayesian model incorrectly predicts a decreasing effect of N on false alarms.

S1 Fig), which allowed for flexible, *N*-dependent decision criteria, to see whether the increasing effect of *N* on false alarms seen in the data might be accounted for by a Bayesian model with suboptimal decision criteria (see S8 and S9 Figs for the fitted decision criteria for each model). On this modified Bayesian model, subjects are calculating the decision variable (log likelihood ratio) in a perfectly Bayesian way, but may have a different decision criterion for each *N*, such that they are generally more permissive of "Feeder present" reports as the total number of pigeons increases. Together, S7 and S8 Figs show that the flexible criterion Bayesian model (A2), among many other models, may correctly capture the effect of *N* on false alarms by adopting a more liberal criterion for declaring "feeder present" as *N* grows larger.

We moreover tested another permissive variant of the Bayesian model which allowed for subjects to have incorrectly internalized the spread of pigeons around the feeder ($\sigma$), and the spread of the feeder around the center of the park ($\sigma_s$). We turned both of these variables into free parameters in model A3 (producing a $\sigma$- and $\sigma_s$- flexible variant of the Bayesian model), but this flexible variant performed similarly poorly to the ideal Bayes model (See Fig 6, model A3).

To further visualize model performance, we next examined "Feeder present" trials in greater detail. Recall that during "feeder present" trials, the generative model was such that any one pigeon had a 50% chance of being "affiliated" with the feeder (i.e., drawn from the feeder-centered Gaussian distribution). As a result, some "feeder present" trials have more affiliated pigeons than others (following a binomial distribution over "feeder present" trials). We would expect trials with a greater number of affiliated pigeons to be more easily assessed as "Feeder

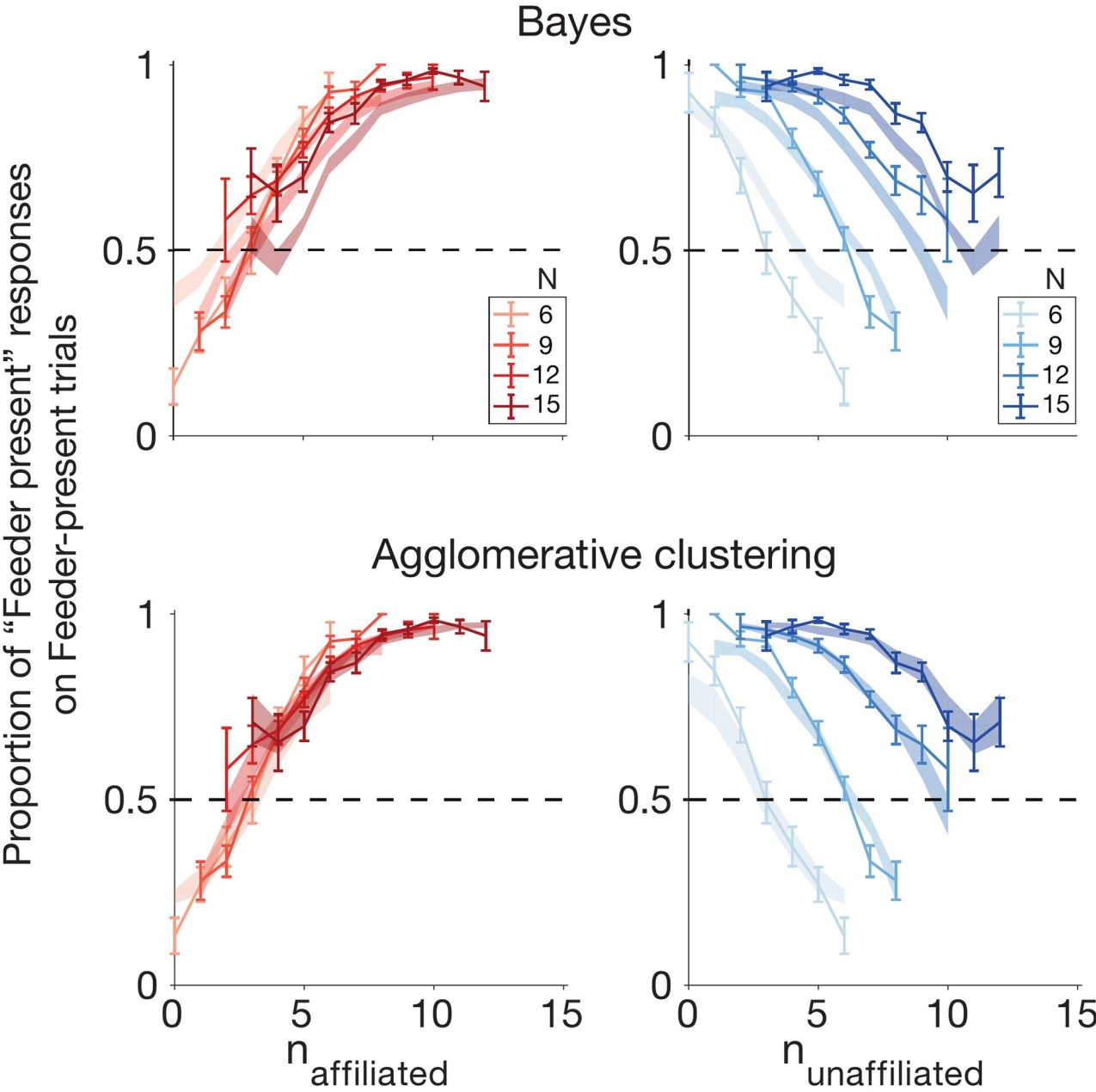

**Fig 5. Subject data (solid) and model fits (shaded), with standard error, showing proportion of "feeder present" responses during "feeder present" trials, as a function of the number of affiliated or unaffiliated pigeons.** Model fits shown for strong Bayes (A1) and Agglomerative Clustering (C8).

present" trials. As expected, for "feeder present" trials, subjects are more likely to correctly respond "feeder present" as the proportion of affiliated pigeons increases, and are more likely to incorrectly respond "feeder absent" as the proportion of unaffiliated pigeons increases (Fig 5). We show that the Strong Bayesian model manages to capture these general trends, but that it provides a rather crude fit to the data. The modified Bayesian model with $N$-dependent decision criteria better captures these data. (S12 Fig)

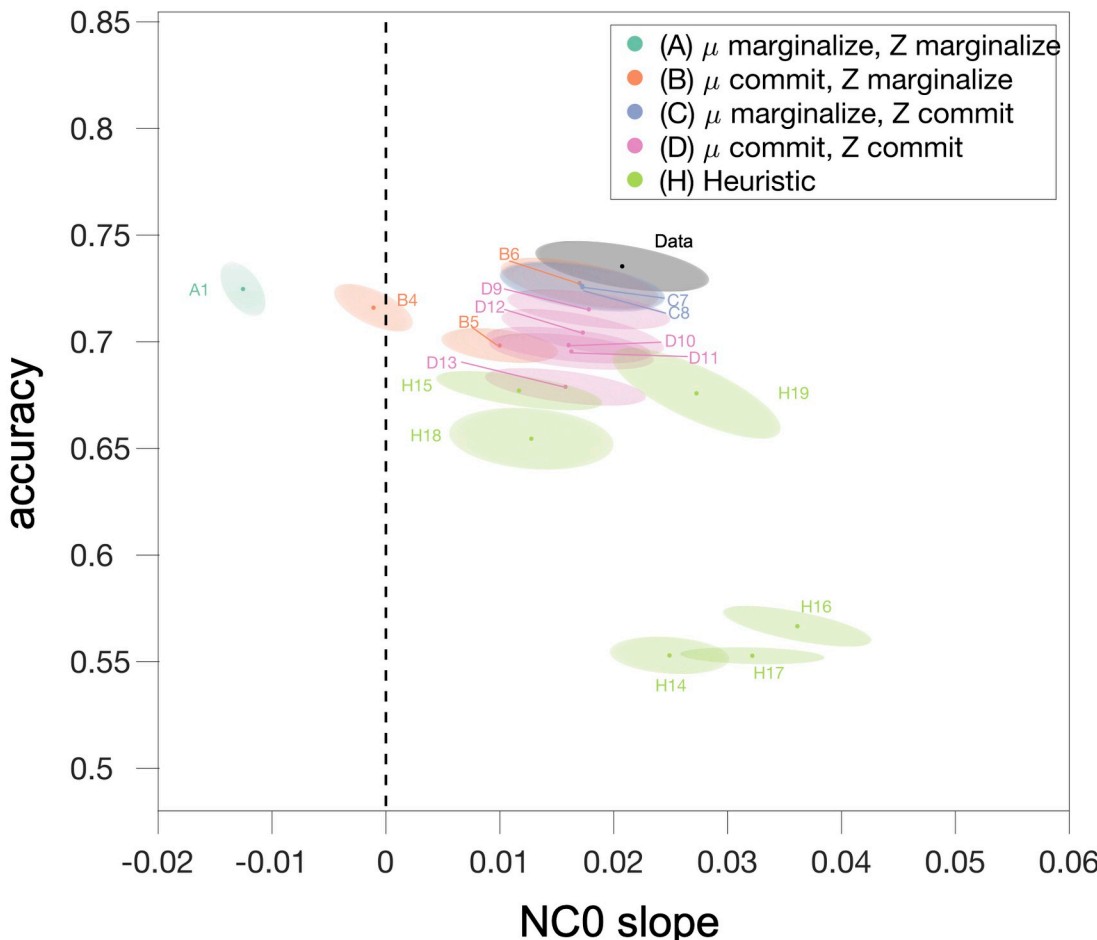

**Fig 6. Two informative dimensions of model predictions.** Overall predicted subject accuracy, and effect of N on false alarms (the slope of the blue shaded line in S7 Fig.

In sum, a Bayesian model with flexible, *N*-dependent criterion (A2) captures subject data well on a number of dimensions. Nevertheless, it still provides an implausible process-level account of the cognitive process, as it requires marginalizing over $2^N$ distinct representations. We next asked whether other models which do not require this taxing marginalization over both variables of the generative model (Families B, C, and D) are able to fit subject data as well as model A2.

To summarize performance over the full range of models, we show two of the most informative aspects of model performance: the model's predicted effect of *N* on false alarms, and the overall predicted subject accuracy (% correct) in Fig 6. Notably, a number of partially committal models (those of Families B, C, and D) perform just as well as the modified Bayesian model A2 on these dimensions. While the ideal Bayesian model A1 predicts subject accuracy reasonably well, it is one of the only that predicts the wrong direction of the effect of *N* on false alarms. Family H models may predict the correct direction of the effect of *N* on false alarms, but are much farther afield in terms of predictions of overall accuracy.

Formal model comparison using Akaike Information Criterion (Fig 7) shows that each of the three "Weak Bayesian" families (B, C, and D) present at least one strong contender for the model that best fits subject data: B6 (in which $\mu$ is committed to via estimate of the posterior

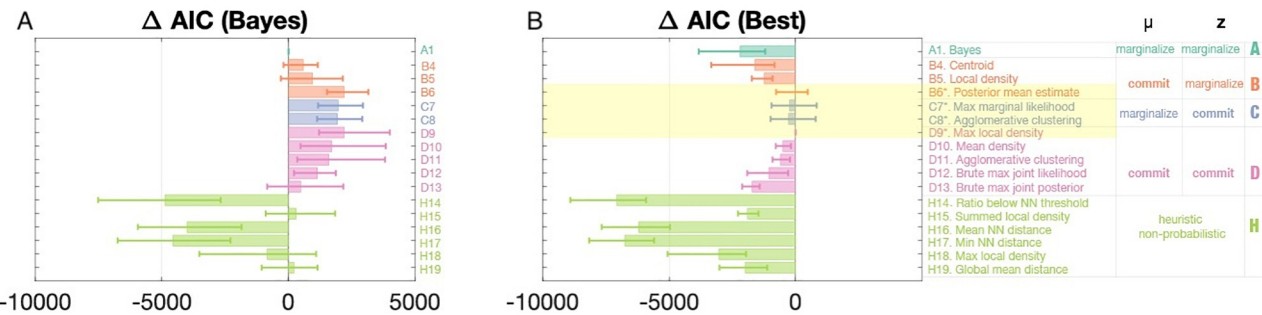

**Fig 7. Model comparison (AIC).** MML denotes "Maximum Marginal Likelihood," MMP denotes "Maximum Marginal Posterior," MJP denotes "Maximum Joint Posterior." See NN denotes "Nearest Neighbour." See "Model Descriptions" under "Methods" for definitions of each model. AIC (Akaike information criterion) is an estimated score for the quality of each model, and the difference in AIC scores between models (Δ AIC) plotted provides an estimate of the quality of each model relative to the fully Bayesian model (A1) in panel A, or the best-fitting model (D9) in panel B. The four best models (highlighted) are indistinguishable via AIC and BIC (see S3 Fig).

mean), C7 (in which **z** is committed to via maximizing the marginal likelihood), C8 (in which **z** is committed to via agglomerative clustering), and D9 (in which both $\mu$ and **z** are jointly committed to via maximizing the joint posterior). These results hold for model comparison via Bayesian Information Criterion (see S3 Fig).

Lastly, it's possible that subjects have not properly learned that the probability of affiliation of a given pigeon is 0.5, and instead hold the false belief that probability of affiliation is *N*-dependent. We tested family A, B, C, and D models again but allowed for probability of affiliation to vary by *N* (See Fig 8 for model comparison, S1 Fig for model parameters, and S11 Fig for fitted $p_{aff}$ values for each *N* for each model). For instance, subjects may have a different internalized probability of affiliation for each *N*, such that they generally believe the feeder to have a higher probability of producing affiliated pigeons when the number of pigeons decreases. This might similarly work to account for the effect of *N* on false alarms: it would mean that subjects require higher clustering to declare "Feeder present" on low-*N* trials, because "Feeder present" as the total number of pigeons increases. We see that with this variation, the Bayesian model can be rescued (see Fig 8 model A1 vs. A2 and S12 Fig model A1P vs.

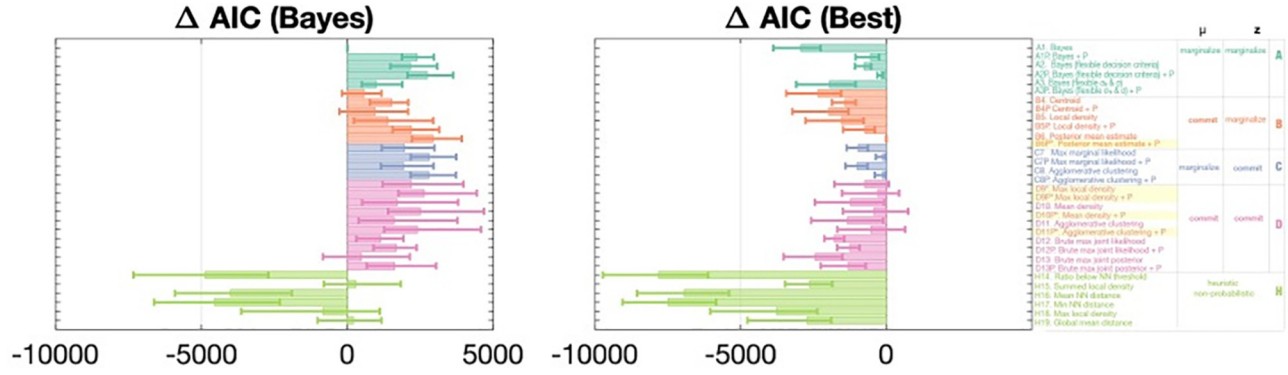

**Fig 8. Model comparison (AIC) with probability of affiliation fitted as a separate free parameter for each of the four *N* conditions (*N* = 6, 9, 12, 15) are denoted with the addition of "P" to the model name.** MML denotes "Maximum Marginal Likelihood," MMP denotes "Maximum Marginal Posterior," MJP denotes "Maximum Joint Posterior." NN denotes "Nearest Neighbour." See "Model Descriptions" under "Methods" for definitions of each model. AIC (Akaike information criterion) is an estimated score for the quality of each model, and the difference in AIC scores between models (Δ AIC) plotted provides an estimate of the quality of each model relative to the fully Bayesian model (A1) in panel A, or the best-fitting model (B6P) in panel B. The five best models (highlighted) are indistinguishable via AIC and BIC (see S4 Fig) Adding a flexible *N*-dependent criterion can help rescue the Bayesian model (see model A2 compared to A1), and adding 4 flexible *N*-dependent probability of affiliation parameters increases goodness of fit for all models. See S1 Fig for more information about the parameters included in each model.

A2P). A number of family B and D models also become indistinguishably well-fitting with the additional $N$-dependent free $p_{aff}$ parameters. On this assumption of false belief, models B6P, D9P, 10DP, and 11DP marginally outperform A2P, C7P, and C8P.

Among the basic (non-false-belief) set of winning models, Agglomerative clustering (C8) is the only one of the four winning models that evades the combinatorially explosive marginalization step over **z** while also providing a potential process-level model of mental representation in the brain, and may be able to account for choice, reaction time, and confidence rating data simultaneously. Fig 9 and S5 Fig show that the approximate log likelihood ratio given by the agglomerative clustering algorithm tracks both reaction times and confidence ratings in a similar manner to that of the Bayes-optimal log likelihood ratio, and Fig 9 suggests that the number of iterations performed by the agglomerative clustering algorithm is comparable to the reaction times of participants for trials binned by agglomerative clustering approximated LLRs. Importantly, Figs 5 and 4 demonstrate that the agglomerative clustering model is able to provide a superior fit to response data compared to the Bayesian model (A1) and an indistinguishably good fit to the other three basic winning models (Fig 7).

In Agglomerative Clustering, the observer commits to a particular **z** rather than representing a full probability distribution. We choose this **z** by picking a single point as the cluster "seed." The cluster is hypothesized to belong to the causal source. One iteratively adds the next-nearest point to the cluster, each time evaluating the log likelihood ratio which results from that particular **z**. We continue adding points until the log likelihood ratio no longer increases, resulting in some spatially contiguous set of points hypothesized to belong to the source if such a source exists, represented by the committed **z**. The **z**-dependent log likelihood ratio is then calculated as a decision variable with noisy threshold. This algorithm is loosely based on work by Heller and Gharamani on agglomerative hierarchical clustering [17]. Our algorithm is sequential rather than hierarchical, in that a single cluster is defined and single observations are merged to the existing cluster via Bayesian hypothesis testing (i.e., the

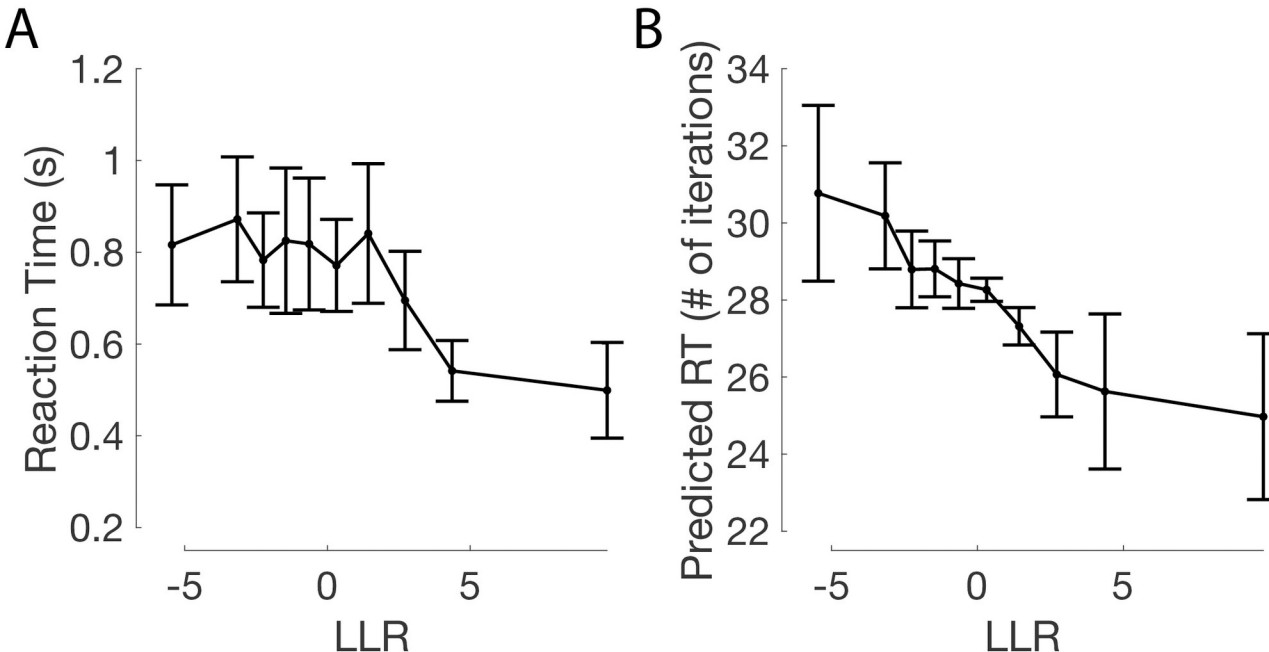

**Fig 9.** (A) Subject reaction times and (B) Predicted reaction times (based on the number of iterations of the agglomerative clustering algorithm), according to the log likelihood ratios (LLR) of the Bayesian model, binned into 10 LLR quantiles (equal number of trials in each plot point).

evaluation of the log likelihood ratio). For a histogram comparing the Bayesian LLR to the Agglomerative Clustering approximation of the LLR, see Fig 10.

S8 Fig shows fitted decision criteria for each model across subjects. There are a number of factors that contribute to the fitted decision criterion across models– for instance, the

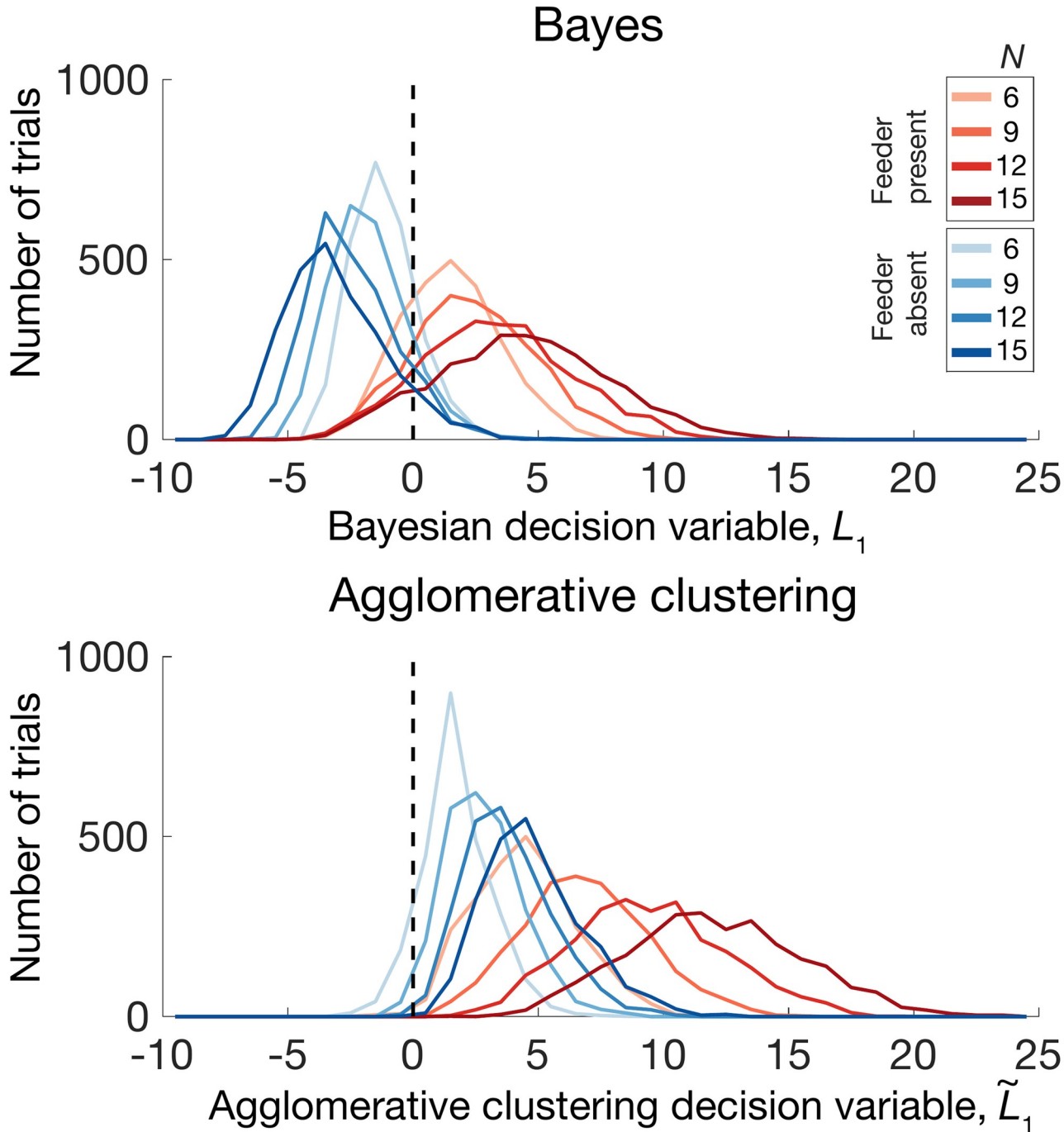

**Fig 10. Histogram of decision variables across trials, pooled across all subjects (2000 trials x 10 subjects, plotted in 35 equal bins from -10 to 25).** The decision variable is the log likelihood ratio for Bayes (A1), and an estimated log likelihood ratio for agglomerative clustering (C8), calculated based on the stimulus on each trial. Note that as the number of pigeons increases during feeder-absent trials, the distribution of the decision variable for agglomerative clustering shifts to the right (light blue to dark blue lines), whereas the Bayesian decision variable exhibits the opposite trend.

goodness of the committed variables, and the subject's fitted $p_{aff}$ beliefs. Some models commit to a particular $\mu$ or $z$ that is particularly good, thereby biasing the log posterior ratio estimates to higher values. Overall positive decision criteria might therefore be expected if the committed variables are quite good– for example, Model 9 evaluates the LLR at only the $\mu$ and $z$ that maximizes the joint posterior, meaning the estimated LLR for a given trial will shift upwards (see S10 Fig, panel A). Similarly, models with $p_{aff} > 0.5$ may cause the decision variable distributions (and therefore the fitted decision criteria) to shift downwards. If the probability of affiliation for the feeder present ($C = 1$) hypothesis is higher than 0.5, then it should take a greater amount of dot "clustering" to support the feeder present hypothesis, and the observer's decision variable on a given trial will shift lower, in the direction of the feeder absent ($C = 0$) hypothesis (see S10 Fig, panel B). Note that the decision variable's zero-point is only meaningful for the perfect Bayesian model (since a Bayesian observer should set their decision criterion to zero), whereas observers likely set a different decision threshold when the log posterior ratio is estimated (as in all other models), in compensation for the fact that these estimates produce shifts in both $C = 0$ and $C = 1$ distributions.

Decision variables (ex, the Bayesian log likelihood ratio, or suboptimal estimates of it) largely tracked confidence ratings and reaction times in expected ways. Confidence ratings reliably tracked Bayesian log likelihood ratios in a monotonic manner (S5 Fig). High-confidence "Feeder present" responses corresponded to high positive log likelihood ratios, while high-confidence "Feeder absent" responses tracked low negative log likelihood ratios. Reports of higher confidence were also related to shorter reaction times (S6 Fig). Trials which fell into the highest positive LLR quantile (highly clustered pigeon configurations which greatly favour "Feeder present" responses) were also related to shorter reaction times.

Notably, Agglomerative Clustering, being the only model with a structured iterative "time course", is able to predict trends in reaction time data (Fig 9). When binning stimuli based on their estimated LLRs, trends in the number of iterations required of the agglomerative clustering algorithm mirror trends in subject reaction times, as seen in (Fig 9. For most subjects (8 out of 10), there is a very weak but significant ($p < 0.001$) ranked correlation between the reaction times and the number of iterations of the agglomerative clustering algorithm across trials (across subjects, mean Spearman $\rho$ across subjects = 0.12, SEM = 0.02).

## Discussion

In this paper, we asked how people infer the presence or absence of a latent cause based on a spatial configuration of items, especially when optimal inference would require the integration of a computationally intractable number of world states. We first asked whether observer models that make inferences over aspects of the true generative model (i.e., Bayesian models and point-estimating variants) accounted better for human behaviour than models based on simple visual heuristics, such as mean density or minimal distance. We found that this was the case, with the heuristic models deviating greatly from the data. We fitted three families of partially committal or point-estimating models, in each of which either or both of the nuisance variables of the generative model were represented as point estimates rather than probability distributions. We found that the Bayesian model (which represents both generative model variables as full probability distributions) exhibits qualitative deviations from subject behaviour. Notably, the Bayesian model fails to capture the trend that observers are more likely to falsely detect a latent cause as the number of visual elements increased. Several point estimate observer models—at least one from each family—can account for these deviations, as well as for fine-grained subject behaviour in its entirety. Therefore, point estimate representation of certain generative model variables may be how the brain performs causal inference, especially when full marginalization over these variables requires accounting for a large number of world states.

In context of this complex latent cause detection task, the effect of the number of observations on false alarm rates was strong. Recent work tested the ability of subjects to infer a hidden low-level variable (the proportion of red vs. blue team supporters leaving a given airplane) based on both observations of differently sized samples, and an inferred higher-level contextual variable (the general red or blue-team bias for a set of planes). [18] show that inferred context is integrated with observations to inform confidence estimate, and notably that the sample size of previous observations guides context reliability. The sample size of observations may therefore be a factor that subjects take into consideration when performing hierarchical latent variable inference in the real world, affecting both choice and confidence judgements. In the case of our study, however, subjects suboptimally set a more liberal detection criterion as sample size increased, perhaps reflecting certain real-world prior beliefs that a greater number of observations provides evidence for the presence of a latent spatial cause, regardless of the distribution of the observations.

Among the basic set of models (that do not make false belief assumptions about the probability of feeder affiliation), the four best-performing models (B6, C7, C8, D9) were functionally indistinguishable according to AIC and BIC. However, taking seriously the computations required of some of these winning models at the algorithmic level would still mean granting the brain upwards of $2^N$ evaluations in the case of marginalization over $\mathbf{z}$. The agglomerative clustering account (C8) stands out because the number of operations required to commit to a plausible partition is less than $\mathcal{O}(N^2)$, making it a more attractive candidate as a biologically plausible algorithm compared to any of the other three statistically indistinguishable partially Bayesian models. The algorithm further coheres with intuitions that the least plausible possibilities of $\mathbf{z}$ may not be represented or considered at all during the decision-making process. The iterative nature of the agglomerative clustering algorithm moreover allows it to predict response times, whereas the Bayesian model and variants thereof do not provide an immediate mechanism to do so.

However, the present study only reflects a small sampling of the possible models in each tested family. This means that we have to be cautious when making any inferences about which generative model variables may be committed to as a point estimate. The study is also limited in its ability to provide an explanation for why the brain may represent certain generative model variables as point estimates while representing others as full probability distributions. Nevertheless, the models tested represent a new and more or less principled category of approximately Bayesian algorithms, distinct from sampling and variational algorithms.

These results advance an understanding of perceptual grouping and causal inference based on spatially-related image elements, suggesting that, while observers likely represent the structure of the generative model, they probably do not represent full probability distributions and marginalize over each nuisance variable. Committing to the representation of some nuisance variables as point estimates may be one strategy employed by the brain to avoid computationally costly marginalizations in real-world visual environments, if these marginalizations are in fact shown to be costly to the brain. Though the present study cannot shed light on why some variables may be represented as full probability distributions but not others, testing algorithms that implement these partially committal models may be a fruitful inductive modelling approach for other perceptual decision-making tasks, as they may help reformulate intractable "as if" Bayesian computations in terms of a set of tractable process-level algorithms potentially used by the brain.

## Methods

### Ethics statement

This experiment was approved by the University Committee on Activities Involving Human Subjects of New York University (IRB-FY2019–2490). All subjects were naive to the purpose

of the experiment and written informed consent was given by each subject at the start of the first session.

## Subjects

10 subjects (6 male), aged 19–39, completed the experiment in two 30–45 minute sessions spaced at least 4 hours apart. Subjects received $12 per session and a $12 completion bonus.

## Apparatus

Subjects were seated in a well-lit room, and stimuli were displayed using a 13.3-inch (1440 x 900) MacBook Air (early 2015 model, Intel HD Graphics 6000 1536 MB). Responses were given on the laptop keyboard. The experiment was run using the Psychtoolbox-3 (V3.0.14) package for MATLAB 9.2.

## Stimuli

The background for all displays was black. Stimulus dots (denoting pigeon locations) were white. Dots were displayed within a circular arena measuring 7 cm in radius positioned at the center of the laptop screen. The circular perimeter of the arena was marked with a thin gray line.

## Procedure

**Trial structure.** Subjects completed two 30–45 minute sessions, spaced at least 4 hours apart. The first session began with a detailed set of instructions delivered verbally and repeated again in writing before the start of each session. Each subject completed 2000 trials in total, split evenly between the two sessions. Each session contained 5 blocks (200 trials per block), and lasted approximately 6 minutes (approximately 1.8 seconds per trial).

**Instructions and category training.** At the beginning of the first session, subjects completed a 10-minute demo which explained the task with visual examples and provided 10 practice trials on which full feedback is given. Subjects scrolled through the tutorial at their own pace. The demo first introduced, in writing and with visuals, the boundaries of the park (the circle), the center of the park (denoted by a grey cross), and the pigeons (denoted by white dots). It was next explained that "On HALF (50%) of the days, there is no pigeon feeder. This means pigeons were scattered randomly within the circle. . ." and that "With NO FEEDER, the pigeons are all UNAFFILIATED PIGEONS." This was followed by 10 randomly drawn examples of pigeon configurations with no pigeon feeder present (C = 0).

Next, the pigeon feeder was introduced (denoted by a dot of a different colour). The pigeon feeder was introduced as ". . . a lonely old ghost who appears in the park on 50% of the days. She typically appears near the center of the park, but she tends to stray." 10 randomly drawn examples were then given of the pigeon feeder's location. It was explained that on exactly "50%" of all days, the pigeon feeder visits the park, but cannot be observed directly. Lastly, the pigeon distribution on "feeder present" days was explained and demonstrated: "When the pigeon feeder is present, pigeons tend to cluster around her location. But even pigeons can't always sense her presence. Even when the feeder is present, there's only a 50% chance that a given pigeon will be AFFILIATED with her." This was then followed by 10 examples of the park when the feeder was present. On these displays, subjects were instructed to toggle the space bar to reveal the true location of the feeder, as well as to reveal the pigeons which were affiliated vs. unaffiliated with the feeder (these dots appear in different colours on the screen). Subjects were then instructed to tell apart the days on which the ghostly pigeon feeder is absent

(left-hand) vs. present (right-hand), and were instructed to use the full range of confidence values when they respond.

The tutorial was followed by 20 practice trials on which full feedback was given– the response was marked as correct or incorrect, and for "feeder present" trials, the location of the feeder as well as the unaffiliated and affiliated pigeons were revealed. These practice trials were not included in the main analysis. See github.com/jennlauralee/pigeons to run the full demo on MATLAB (requires Psychtoolbox).

**Testing.** At the beginning of each testing block, instructions were displayed on the screen which show the mapping of keyboard buttons to confidence and category responses. At the end of each trial, subjects received partial feedback about whether their categorization was correct or incorrect (ex, "Incorrect! You reported absent. The feeder was present.") Subjects completed 200 trials per block, with five blocks per session. Subjects were able to take breaks between blocks.

## Model derivations

Below, we introduce the notation used for all models and formalize the true generative model that determines how observations are in fact generated based on stimulus category. Then, we formalize each of the observer models within the five observer model families, each of which perform full (family A) or partial (families B, C, D) generative model inferences about stimulus category based on observations, or which simply employ a heuristic decision model (family H).

## Notation

All location variables have two dimensions, but we do not denote them as vectors (with boldface). We reserve boldface notation for vectors corresponding to $N$ observations.

| | |
|---|---|
| $C$ | Latent cause variable (0 if absent, 1 if present) |
| $p_{\text{aff}}$ | Probability that an observation is drawn from the latent cause, when $C = 1$ ($p_{\text{aff}} = 0.5$ in the experiment) |
| $R$ | Radius of disc from which each "background" (non-latent-cause) observation is drawn |
| $\mu$ | Location of latent cause |
| $\sigma_{\text{s}}$ | Standard deviation of circular Gaussian distribution with mean $\mathbf{0}$ from which $\mu$ is drawn |
| $x_i$ | $i^{\text{th}}$ observation |
| $\mathbf{x}$ | Set of all observations, $\{\mathbf{x}_1, \ldots, \mathbf{x}_N\}$ |
| $\sigma$ | Standard deviation of circular Gaussian distribution with mean $\mu$ from which each latent-cause observation is independently drawn |
| $z_i$ | Binary variable that takes the value 1 if $x_i$ is drawn from the latent cause and 0 if $x_i$ is drawn from the background |
| $\sigma_d$ | Standard deviation of response noise |
| $\lambda$ | Lapse rate |

## Generative model

Here we discuss the generative model that determines how pigeon locations are generated in the circular park based on stimulus category.

**Category.** The experimenter designates each trial randomly as either a "Feeder Present" ($C = 1$) or "Feeder Absent" ($C = 0$) trial. There are an equal number of feeder present and absent trials such that

$$p(C = 0) = p(C = 1) = 0.5 \tag{4}$$

If $C = 0$, the distribution of each observation is uniform on a disc:

$$p(x_i|C=0) = \begin{cases} \dfrac{1}{\pi R^2} & \text{for} ||x_i|| < R \quad (5) \\ 0 & \text{otherwise} \quad\quad (6) \end{cases}$$

If $C = 1$, the pigeon feeder is present, and its location $\mu$ is drawn from a Gaussian centered at the middle of the circular park (position (0,0)) with standard deviation $\sigma_s$.

$$p(\mu|C=1) = \mathcal{N}(\mu; (0,0), \sigma_s^2 I), \tag{7}$$

where $I$ is the two-dimensional identity matrix. For a given $\mu$, the pigeon locations are independent

$$p(\mathbf{x}|\mu, C=1) = \prod_{i=1}^{N} p(x_i|\mu, C=1) \tag{8}$$

For each of $N$ pigeons drawn on a given trial, there is a $p_{\text{aff}}$ chance that the pigeon is affiliated with the pigeon feeder (in the experiment, $p_{\text{aff}} = 0.5$). If a pigeon is not affiliated ($z_i = 0$), its location is drawn from a uniform distribution on a disc of radius $R$ centered on the center of the screen:

$$p(x_i|z_i=0, C=1) = \begin{cases} \dfrac{1}{\pi R^2} & \text{for} ||x_i|| < R \quad (9) \\ 0 & \text{otherwise} \quad\quad (10) \end{cases}$$

If the pigeon is affiliated with the feeder ($z_i = 1$), then the pigeon's location is drawn from a smaller Gaussian distribution centered at the location of the pigeon feeder ($\mu$) with standard deviation $\sigma$:

$$p(x_i|z_i=1, \mu, C=1) = \mathcal{N}(x_i; \mu, \sigma^2 \mathbf{I}) \tag{11}$$

This means the distribution of pigeon observations is a mixture of a Gaussian distribution centered at $\mu$ and a uniform distribution on the same disc:

$$p(x_i|\mu, C=1) = \begin{cases} p_{\text{aff}}\mathcal{N}(x_i; \mu, \sigma^2 I) + \dfrac{1 - p_{\text{aff}}}{\pi R^2} & \text{for} ||x_i|| < R \quad (12) \\ \mathcal{N}(x_i; \mu, \sigma^2 I) & \text{otherwise.} \quad\quad (13) \end{cases}$$

Through our choice of $\sigma_s$ and $\sigma$, the latter case where $||x_i|| > R$, in which the location of a generated pigeon falls outside of the perimeter of the circular park, rarely occurs. We ignore this case moving forward.

## Observer models

Next, we discuss observer models that assume some knowledge of the generative model and invert the model to perform inference about the trial's category (feeder absent or present) based on pigeon location observations.

**Bayes-optimal model.** Given $N$ observations $\mathbf{x} = \{x_1, \ldots, x_N\}$, the observer is tasked to infer category $C$. The log posterior ratio is

$$d = \log\frac{p(C=1|\mathbf{x})}{p(C=0|\mathbf{x})} = \log\frac{p(C=1)}{p(C=0)} + \log\frac{p(\mathbf{x}|C=1)}{p(\mathbf{x}|C=0)} \tag{14}$$

Since $p(C = 0) = p(C = 1) = 0.5$, we have

$$d = \log \frac{L_1}{L_0},\tag{15}$$

where we have introduced the following notation for the category likelihoods:

$$L_0 \equiv p(\mathbf{x}|C = 0);\tag{16}$$

$$L_1 \equiv p(\mathbf{x}|C = 1).\tag{17}$$

Before we evaluate these expressions further, we first comment on how the observer would make a decision based on $d$. An observer who maximizes probability correct will report $C = 1$ when $d > 0$. However, an observer may have utilities or costs that are unequal between $C = 0$ and $C = 1$, such as unequal degrees of satisfaction for correctly reporting the presence of the latent cause than for correctly reporting its absence, or a higher motor cost for pressing a button with the non-preferred hand (in our experiment, hands corresponded one-to-one to category responses). Due to such factors, which we do not model explicitly, the optimal Bayesian decision-maker might use a decision criterion that is different from 0 and might even depend on the number of observations. Since criterion-setting is not the subject of our study, we remain as agnostic as possible about the criteria: we assume that the Bayesian observer reports $C = 1$ when $d > k$, where $k$ is fit as a free parameter. Incidentally, this also allows for a mismatched prior over $C$, although this would be a deviation from strictly optimal decision-making.

In addition, we allow for decision noise, which we model as Gaussian noise on the log posterior ratio $d$, with a variance that we denote by $\sigma_d$. Finally, we allow for a lapse rate $\lambda$; this is the probability of randomly choosing either category with probability 0.5. Putting everything together, the probability of reporting $C = 1$ becomes

$$p(\hat{C} = 1|\mathbf{x}) = 0.5\lambda + (1 - \lambda)\Phi\left(\log \frac{L_1}{L_0}; k, \sigma_d^2\right)\tag{18}$$

We now return to the calculation of the likelihoods of both categories, from Eq (17). The likelihood of the hypothesis $C = 0$ is simply

$$L_0 = \frac{1}{(\pi R^2)^N},\tag{19}$$

whereas the likelihood of the hypothesis $C = 1$ is obtained by marginalizing over all variables in the generative model other than $C$ and $\mathbf{x}$, that is, over $\mathbf{z}$ and $\mu$:

$$
\begin{aligned}
L_1 &= p(\mathbf{x}|C = 1) && (20)\\
&= \sum_{\mathbf{z}} \int p(\mathbf{x}|\mathbf{z}, \mu, C = 1)p(\mathbf{z})p(\mu)d\mu, && (21)\\
&= \sum_{\mathbf{z}} \int L(\mathbf{z}, \mu)p(\mathbf{z})p(\mu)d\mu, && (22)
\end{aligned}
$$

where $p(\mathbf{z})$ is the probability of the vector of affiliations and $L(\mathbf{z}, \mu) \equiv p(\mathbf{x}|\mathbf{z}, \mu, C = 1)$ is the

joint likelihood of the combination $(\mathbf{z}, \mu)$ under the $C = 1$ hypothesis. The former is equal to

$$p(\mathbf{z}) = (1 - p_{\text{aff}})^{N_0(\mathbf{z})} p_{\text{aff}}^{N_1(\mathbf{z})} \tag{23}$$

where $N_0$ and $N_1$ are the number of unaffiliated and affiliated pigeons.

We further evaluate the joint likelihood by using the property that the pigeon locations are drawn independently, that the location of an unaffiliated pigeon ($z_i = 0$) is drawn from a uniform distribution on a disc, and that the location of an affiliated pigeon ($z_i = 1$) is drawn from a normal distribution with mean $\mu$ and standard deviation $\sigma$:

$$L(\mathbf{z}, \mu) = \prod_{i=1}^{N} p(x_i|z_i, \mu, C = 1) \tag{24}$$

$$= \left(\frac{1}{\pi R^2}\right)^{N_0(\mathbf{z})} \left(\prod_{i:z_i=1} p(x_i|z_i = 1, \mu, C = 1)\right) \tag{25}$$

$$= \left(\frac{1}{\pi R^2}\right)^{N_0(\mathbf{z})} \left(\prod_{i:z_i=1} \mathcal{N}(x_i; \mu, \sigma^2 \mathbf{I})\right) \tag{26}$$

We can write $L_1$ in two equivalent "two-step" ways. The first is in terms of a likelihood over $\mathbf{z}$:

$$L_1 = \sum_{\mathbf{z}} L(\mathbf{z}) p(\mathbf{z}) \tag{27}$$

$$L(\mathbf{z}) \equiv \int L(\mathbf{z}, \mu) p(\mu) d\mu \tag{28}$$

$$= p(\mathbf{x}|\mathbf{z}, C = 1). \tag{29}$$

The second is in terms of a likelihood over $\mu$:

$$L_1 = \int L(\mu) p(\mu) d\mu, \tag{30}$$

$$L(\mu) \equiv \sum_{\mathbf{z}} L(\mathbf{z}, \mu) p(\mathbf{z}) \tag{31}$$

$$= p(\mathbf{x}|\mu, C = 1). \tag{32}$$

Here, $L(\mu)$ can be written in multiple equivalent ways:

$$L(\mu) = \sum_{\mathbf{z}} p(\mathbf{z}) \prod_{i=1}^{N} (p(x_i|z_i, \mu, C = 1)) \tag{33}$$

$$= \sum_{\mathbf{z}} p(\mathbf{z}) \left(\frac{1}{\pi R^2}\right)^{N_0(\mathbf{z})} \prod_{x_i:z_i=1} \mathcal{N}(x_i; \mu, \sigma^2 \mathbf{I}) \tag{34}$$

$$L(\mu) = \prod_{i=1}^{N} \sum_{z_i=0}^{1} (p(x_i|z_i, \mu, C=1)p(z_i)) \tag{35}$$

$$= \prod_{i=1}^{N} \left( \frac{1 - p_{\text{aff}}}{\pi R^2} + p_{\text{aff}} \mathcal{N}(x_i; \mu, \sigma^2 \mathbf{I}) \right) \tag{36}$$

For all suboptimal models, we consider different ways of approximating the evidence for the causal model (i.e. the "feeder persent" likelihood). The notation for this approximation is $\tilde{L}_1$.

## Model descriptions

**Family A—Marginalize over both $\mu$ and $z$.** Family A models include the Bayes-optimal model and two variants that allow for false beliefs about the generative model.

A(1) Strong Bayes-optimal model

The strong Bayes-optimal model is as described above. $p_{\text{aff}} = 0.5$ in all default models, but an $N$-dependent flexible $p_{\text{aff}}$ variant of most models is also fit, to reflect the hypothesis that subjects may have different baseline priors or propensities to report $C = 1$ as the number of pigeons $N$ increases.

A(2) Bayes-optimal model with N-dependent decision criteria

A variant of the Bayesian model where a separate decision criterion $K_N$ is fit for each of $N = 6, 9, 12, 15$ pigeons. Model A1 reflects a strong Bayesian model with one fitted decision criterion, while model A2 reflects a modification of the Bayesian model with four decision criteria fitted per subject. We equip all alternative models with the same N-dependent flexibility in criteria moving forward.

A(3) Bayes-optimal model with flexible beliefs about $\sigma$ and $\sigma_s$

A variant of the Bayesian model where we test false beliefs about the generative model by fitting $\sigma$ and $\sigma_s$ as free parameters.

**Family B—Commit to a single $\mu$, marginalize over $z$.** Family B models employ the strategy of committing to a single hypothesized feeder location $\hat{\mu}$, then reasoning probabilistically over all possible combinations of each point having been generated from the feeder distribution or the uniform background distribution. The general form of the suboptimality is

$$\begin{aligned}
\tilde{L}_1 &= L(\hat{\mu}; \mathbf{x}) \\
&= \sum_{\mathbf{z}} L(\mathbf{z}, \hat{\mu}; \mathbf{x}) p(\mathbf{z}) \\
&= \prod_{i=1}^{N} p(x_i|\hat{\mu})
\end{aligned}$$

We tested 4 possible ways of arriving at $\hat{\mu}$. The first two benefit from easy heuristic interpretations, while the second two involve cognitively implausible computations.

B(4) Commit to $\mu$ as location of centroid

$$\hat{\mu} = \frac{\Sigma(\mathbf{x})}{N}$$

B(5) Commit to $\mu$ as location of maximum local density

- A Gaussian with $\sigma = 1.4$ cm (reflecting the statistics of bird dispersal around the bird feeder) is convolved along all possible locations of the arena, as a measure of the local density of dots at all possible locations on the screen.

- The location of the highest local density is taken as $\hat{\mu}$.

B(6) Commit to $\mu$ through posterior mean estimation

$$\hat{\mu} = \int \mu \, p(\mu|\mathbf{x}, C = 1) d\mu \tag{37}$$

The posterior over $\mu$ is obtained by multiplying the prior over the distribution of $\mu$ by the likelihood in Eq (34):

$$p(\mu|\mathbf{x}, C = 1) \propto p(\mu) \sum_{\mathbf{z}} \frac{p(\mathbf{z})}{(\pi R^2)^{N_0(\mathbf{z})}} \prod_{x_i:z_i=1} \mathcal{N}(x_i; \mu, \sigma^2 \mathbf{I}) \tag{38}$$

$$= \sum_{\mathbf{z}} \frac{p(\mathbf{z})}{(\pi R^2)^{N_0(\mathbf{z})}} p(\mu) \prod_{x_i:z_i=1} \mathcal{N}(x_i; \mu, \sigma^2 \mathbf{I}) \tag{39}$$

$$= \sum_{\mathbf{z}} \frac{p(\mathbf{z})}{(\pi R^2)^{N_0(\mathbf{z})}} \mathcal{N}(\mu; 0, \sigma_\mu^2 \mathbf{I}) \prod_{x_i:z_i=1} \mathcal{N}(x_i; \mu, \sigma^2 \mathbf{I}) \tag{40}$$

This is a mixture of a product of Gaussians, with mixture weights

$$w(\mathbf{z}) \equiv \frac{p(\mathbf{z})}{(\pi R^2)^{N_0(\mathbf{z})}}. \tag{41}$$

Writing the product of Gaussians as a single (unnormalized) Gaussian, we find

$$p(\mu|\mathbf{x}, C = 1) \propto \sum_{\mathbf{z}} w(\mathbf{z}) \mathcal{N}(\mu; 0, \sigma_\mu^2 \mathbf{I}) \prod_{x_i:z_i=1} \mathcal{N}(x_i; \mu, \sigma^2 \mathbf{I}) \tag{42}$$

$$= \sum_{\mathbf{z}} w(\mathbf{z}) \tilde{w}(\mathbf{x}, \mathbf{z}) \mathcal{N}(\mu; m(\mathbf{x}, \mathbf{z}), v(\mathbf{z})), \tag{43}$$

where

$$m(\mathbf{x}, \mathbf{z}) = \frac{x_{\text{sum}}(\mathbf{z})}{N_1(\mathbf{z}) + \frac{\sigma^2}{\sigma_\mu^2}} \tag{44}$$

$$v(\mathbf{z}) = \frac{1}{\frac{1}{\sigma_\mu^2} + \frac{N_1(\mathbf{z})}{\sigma^2}} \mathbf{I} \tag{45}$$

$$\tilde{w}(\mathbf{x}, \mathbf{z}) = \frac{1}{1 + \frac{\sigma_\mu^2}{\sigma^2} N_1(\mathbf{z})} \frac{1}{(2\pi\sigma^2)^{N_1(\mathbf{z})}} \exp\left(-\frac{1}{2\sigma^2}\left[\left(\sum_{x_i:z_i=1} ||x_i||^2\right) - \frac{||x_{\text{sum}}(\mathbf{z})||^2}{N_1(\mathbf{z}) + \frac{\sigma^2}{\sigma_\mu^2}}\right]\right), \tag{46}$$

where $x_{\text{sum}}(\mathbf{z}) = \sum_{x_i:z_i=1} x_i.$

The normalized posterior is

$$p(\mu|\mathbf{x}, C = 1) = \frac{\sum_{\mathbf{z}} w(\mathbf{z})\tilde{w}(\mathbf{x}, \mathbf{z})\mathcal{N}(\mu; m(\mathbf{x}, \mathbf{z}), v(\mathbf{z}))}{\sum_{\mathbf{z}} w(\mathbf{z})\tilde{w}(\mathbf{x}, \mathbf{z})},$$ (47)

and the posterior mean estimate from Eq (37) is

$$\hat{\mu}_{PME} = \frac{\sum_{\mathbf{z}} w(\mathbf{z})\tilde{w}(\mathbf{x}, \mathbf{z})m(\mathbf{x}, \mathbf{z})}{\sum_{\mathbf{z}} w(\mathbf{z})\tilde{w}(\mathbf{x}, \mathbf{z})}.$$ (48)

**Family C—Marginalize over $\mu$, commit to a single $z$.** Family C models employ the strategy of first selecting a group of points which are thought to belong to the feeder distribution. With this partition nailed down, the subject then reasons probabilistically over all possible feeder locations. The general form of the suboptimality is

$$\tilde{L}_1 = L(\hat{\mathbf{z}}; \mathbf{x})$$
$$= \int L(\hat{\mathbf{z}}, \mu; \mathbf{x})p(\mu)d\mu$$

C(8) Commit to $z$ as maximum marginal likelihood

$$\tilde{L}_1 = \max_z L(z; x)$$

C(9) Commit to $z$ via agglomerative clustering

- See agglomerative clustering box (Box 1)

**Family D—Commit to both $\mu$ and z.** General form of the Family D suboptimality is

$$\tilde{L}_1 = L(\hat{\mathbf{z}}, \hat{\mu}; \mathbf{x})$$

D(10) Commit jointly to $\mu$ and $\mathbf{z}$ by maximizing the joint posterior

$$\tilde{L}_1 = \max_{\mathbf{z}, \mu} L(\mathbf{z}, \mu; \mathbf{x})p(\mu)$$

D(11) Commit jointly to $\mu$ and $\mathbf{z}$ by maximizing the joint likelihood

$$\tilde{L}_1 = \max_{\mathbf{z}, \mu} L(\mathbf{z}, \mu; \mathbf{x})$$

D(12) Commit jointly to $\mu$ and $\mathbf{z}$ via agglomerative clustering

- See agglomerative clustering box. (Box 1.)

D(13) Commit to $\mu$ as location of centroid, then commit to $\mathbf{z}$ by setting an optimal radial threshold

D(14) Commit to $\mu$ as location of maximum local density, then commit to $\mathbf{z}$ by setting an optimal radial threshold

> **Box 1**. The Agglomerative Clustering algorithm is as follows:
>
> 1. Pick a starting pigeon at random, to seed the cluster.
>
> 2. Evaluate the log likelihood ratio (LLR): that is, the marginal likelihood that that cluster was generated by the Gaussian (and the rest by the uniform) against the marginal likelihood that all points were generated by the uniform distribution.
>
> 3. Add the next-nearest point to the cluster. Repeat step 2.
>
> 4. If adding that next-nearest point caused the log likelihood ratio to decrease from the previous iteration, stop. The current LLR is the best LLR for this seed. Otherwise, repeat step 3.
>
> 5. Calculate the highest of all best LLRs across all starting cluster seeds (i.e. repeat 1–4 with each pigeon as the starting pigeon).
>
> 6. Use the highest possible LLR as a decision variable (where, if it exceeds some noisy threshold, the subject responds "feeder present").

- For D(13) and D(14): Given an estimate of $\mu$, the ML estimate of $\mathbf{z}$ amounts to applying a specific distance threshold

$$\hat{z}_i = 1 \text{ if } N(x_i, \hat{\mu}, \sigma^2) > \frac{1}{\pi R^2} \Leftrightarrow ||x_i - \hat{\mu}|| < \sqrt{2\sigma^2 \log \frac{R^2}{2\sigma^2}}$$

**Family H—Non-probabilistic heuristic models.** Five prima facie plausible heuristic models were developed based on basic first-pass intuitions about the task.

Each heuristic model has the form

$$p(\hat{C} = 1|x) = \Phi(f(\mathbf{x}); k_n, \sigma_d^2),$$

where $f(\mathbf{x})$ represents one of the following heuristics:

H(10) Negative mean pairwise distance

- The mean pairwise distance of all points is calculated as a measure of global dot density.

- This number is made negative.

- If this metric exceeds a noisy threshold, the model responds "feeder present".

H(11) Maximum local density

- A Gaussian with $\sigma = 1.4$ cm (the true standard deviation of pigeon locations around the feeder) is convolved with a map of the pigeon locations, as a measure of the local density of dots.

- This number is made negative.

- If this local density metric at any one location exceeds a noisy threshold, the model responds "feeder present".

H(12) Negative minimum pairwise distance

- The distance between the two nearest points is calculated.

- This number is made negative.

- If this least distance metric exceeds a noisy threshold, the model responds "feeder present".

H(13) Negative mean nearest-neighbour distance

- The mean distance between all points and their nearest neighbour is calculated.

- This number is made negative.

- If this least distance metric exceeds a noisy threshold, the model responds "feeder present".

H(14) Fraction of points below some nearest-neighbour distance threshold

- The mean distance between all points and their nearest neighbour is calculated.

- This number is made negative.

- If this least distance metric exceeds a noisy threshold, the model responds "feeder present".

We did not design our models to explicitly account for confidence responses. Modelling confidence predictions for each of the 8 possible confidence levels would have required an additional 7 decision boundaries, possibly multiplied by the 4 $N$ conditions, which would have introduced a large number of extra parameters without contributing additional understanding. Therefore, we chose to fit to choices only and subsequently make a zero-parameter prediction for confidence ratings.

## Supporting information

**S1 Fig. List of models and model parameters.** Model parameters. $N$-dependent parameters are denoted by '(4)', reflecting a unique parameter fit for each $N$ = 6, 9, 12, 15 condition. (TIFF)

**S2 Fig. Stimulus histogram distributions along each distance-based heuristic.** Stimulus histogram distributions along each distance-based heuristic. "Feeder absent" trials in blue, "feeder present" trials in red, with colour saturation indicating the number of pigeons on a given trial ($N$ = 6, 9, 12, 15). For each trial, the following quantities were computed: the mean pairwise distance of all pigeons (A), the maximum local density, where local density is computed by convolving a gaussian of $\sigma$ = 1.4cm over the circular arena (B), the nearest neighbour distance, calculated as the mean distance between all pigeons and their nearest neighbour (C), and the distance between the two nearest pigeons (D). (TIFF)

**S3 Fig. Model comparison (Bayesian Information Criterion (BIC)).** Model comparison (Bayesian Information Criterion (BIC)). Compared to AIC, BIC model comparison penalizes model complexity more heavily. MML denotes "Maximum Marginal Likelihood," MMP denotes "Maximum Marginal Posterior," MJP denotes "Maximum Joint Posterior." NN denotes "Nearest Neighbour." The difference in BIC scores between models ($\Delta$ BIC) plotted provides an estimate of the quality of each model relative to the fully Bayesian model (A1) in panel A, or the best-fitting model (D9) in panel B. The four best models (highlighted) are indistinguishable via AIC and BIC. (TIF)

**S4 Fig. Model comparison (Bayesian Information Criterion (BIC)) with probability of affiliation fitted as a separate free parameter.** Model comparison (Bayesian Information Criterion (BIC)) with probability of affiliation fitted as a separate free parameter for each of the

four $N$ conditions ($N$ = 6, 9, 12, 15) are denoted with the addition of "P" to the model name. Compared to AIC, BIC model comparison penalizes model complexity more heavily. MML denotes "Maximum Marginal Likelihood," MMP denotes "Maximum Marginal Posterior," MJP denotes "Maximum Joint Posterior." NN denotes "Nearest Neighbour." The difference in BIC scores between models (Δ BIC) plotted provides an estimate of the quality of each model relative to the fully Bayesian model (A1) in panel A, or the best-fitting model (B6P) in panel B. Adding a flexible $N$-dependent criterion can help rescue the Bayesian model (see model A2 compared to A1), and adding 4 flexible $N$-dependent probability of affiliation parameters increases goodness of fit for all models.
(TIF)

**S5 Fig. Average $d$ (log likelihood ratios) for trials binned by response x confidence pairs across all subjects.** Average $d$ (log likelihood ratios) for trials binned by response x confidence pairs across all subjects. Negative confidence rating denotes "feeder absent" response and positive confidence denotes a "feeder present" response.
(TIFF)

**S6 Fig. Subject reaction times for each response x confidence pair.** Subject reaction times for each response x confidence pair, where negative confidence denotes a "feeder absent" response and positive confidence denotes a "feeder present" response.
(TIFF)

**S7 Fig. Model fits of proportion of "feeder present" responses as a function of number of pigeons.** Model fits of proportion of "feeder present" responses as a function of number of pigeons (N), denoted by shaded area; subject data denoted by solid lines.
(TIFF)

**S8 Fig. Fitted decision criterion parameter $k_N$ for each $N$.** Value of the fitted decision criterion parameter $k_N$ for each $N$, shown for each of the basic models.
(TIFF)

**S9 Fig. Fitted decision criterion parameter $k_N$ for each $N$ for flexible $p_{aff}$ model variants.** Value of the fitted decision criterion parameter $k_N$ for each $N$, shown for each of the model variants with a flexible $p_{aff}$.
(TIFF)

**S10 Fig. Histograms of decision variable distributions for example models.** Histograms of decision variable distributions for the Bayesian model ($p_{aff}$ = 0.5) in the top row, with two examples of shifted decision variable distributions. Panel A: maximum joint posterior (model 9, right-shifted). Panel B: Bayesian model with a false probability of affiliation of 0.7 (left-shifted).
(TIFF)

**S11 Fig. Probabiltiy of affiliation ($p_{aff}$) fits across $N$.** Fitted $p_{aff}$ for each $N$ shown for each model variant.
(TIFF)

**S12 Fig. "Feeder present" responses and model fits.** Model fits of proportion of "feeder present" responses as a function of number of pigeons (N), denoted by shaded area; subject data denoted by solid lines.
(TIFF)

## Author Contributions

**Conceptualization:** Jennifer Laura Lee, Wei Ji Ma.

**Data curation:** Jennifer Laura Lee, Wei Ji Ma.

**Formal analysis:** Jennifer Laura Lee, Wei Ji Ma.

**Funding acquisition:** Wei Ji Ma.

**Investigation:** Jennifer Laura Lee, Wei Ji Ma.

**Methodology:** Jennifer Laura Lee, Wei Ji Ma.

**Project administration:** Wei Ji Ma.

**Resources:** Wei Ji Ma.

**Software:** Jennifer Laura Lee.

**Supervision:** Wei Ji Ma.

**Validation:** Jennifer Laura Lee.

**Visualization:** Jennifer Laura Lee.

**Writing – original draft:** Jennifer Laura Lee, Wei Ji Ma.

**Writing – review & editing:** Jennifer Laura Lee, Wei Ji Ma.

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
