## [Decision Letter · Decision Letter 0]

31 Aug 2021

Dear Ms Lee,

Thank you very much for submitting your manuscript "Point-estimating observer models for latent cause detection" for consideration at PLOS Computational Biology. As with all papers reviewed by the journal, your manuscript was reviewed by members of the editorial board and by several independent reviewers. The reviewers appreciated the attention to an important topic. Based on the reviews, we are likely to accept this manuscript for publication, providing that you modify the manuscript according to the review recommendations.

Sincerely,

Jean Daunizeau

Associate Editor

PLOS Computational Biology

Samuel Gershman

Deputy Editor

PLOS Computational Biology

[LINK]

Reviewer's Responses to Questions

**Comments to the Authors:**

Reviewer #1: The authors introduce and describe a novel experimental paradigm for inferring the presence of a ‘food feeder in a park’ where some nuisance variables (location of the feeder and pigeon association) make exact inference intractable. Then, the authors compare both Bayesian and point-estimating models using model comparison. It looks that some models involving marginalization over the latent variables cannot be rejected based on the choice data alone, so I think that some of the statements (such as the title) could be a little bit interpretative and not directly supported by the data. Nevertheless, the work is very strong in comparing models that marginalize or make assumptions to simplify computations, and thus it is of high value.

It seems that perfect Bayesian estimation is contrasted with point estimation. Another way to simplify marginalization is based on the use of priors that highly constraints the size of the sum to the carried. If my reading is correct, no model in the paper uses some form of priors, and it might be unclear why they are not used within the general context of testing multiple models simultaneously.

Traditionally, the London bombing problem has been treated in a different way, in my understanding: one looks at the distribution of counts per bin, and tests whether the distribution is Poisson or not. Would something like this (which is a classical test) be something that participants might be using?

The effects of N in Fig. 3 are very strong and relevant. Recent work by Schustek and Moreno-Bote (Nat Communications, 2019) show also a very strong effect of sample size N on responses and confidence reports, thus supporting the notion that subjects do not use too simple heuristics to solve hard inference problems. This paper could be commented in the discussion.

Related to the above, the sample size used goes up to N=16, well above the subitizing regime, but models do not seem to incorporate the possibility of numerosity estimation errors.

The agglomerative clustering algorithms looks a good model candidate due to its simplicity (although other models that require marginalization are as good as this one, as the authors indicate). This model seems to be sequential in the way clusters are built, which means that studying eye movement could be a very relevant direction to be explored. Is not model comparison rather limited by the use of choice responses, with no RTs, gazes or confidence estimates?

Reviewer #2: In this paper the authors provide a very thorough set of analyses and comparisons to test a range of models of the computations that might be going on within subjects in a cognitive task. The task is a great example of the sorts of computations all animals must do in every day life, while being simple enough that the potential solutions can be analyzed. The work of the authors is very rigorous, perhaps to the point that so many possibilities are being compared that the main point/thread can be lost amidst the large number of results. Therefore, since I think the work is valuable, my suggestions/critiques revolve around increasing readability, in particular, by highlighting and explaining the important findings more.

Specific comments:

Given the number of results and figures I think it is important they are cited clearly in the text in order, alongside a sentence giving the main finding of the figure. For example I do not see Figures 5 or 6 cited in the text and others appear to be cited out of the numerical order in which they appear.

Some of the figures, specifically Figs 8 and 9 along with the corresponding styles in the supplement have so many lines that the text is hard to read and to scan across to the corresponding curve. I suggest a bit more spacing between rows and clearer – probably larger – fonts are needed.

p.2 the sentence explaining nuisance variables would benefit from being broken down and expanded. E.g. What is the “variable of interest” (an example would be nice – I’m still not sure what it means in the actual experiment, presumably the binary variable “C”). It would also not hurt to add a sentence explaining how a “generative model” is not just any model, since it is a bit of a buzzword/phrase these days.

p.7 “following a bimodal distribution” – please explain why “feeder present trials” would have a bimodal distribution of the numbers of affiliated pigeons, since the process appears to be Binomial which has a unimodal distribution.

p.8 “one strong contender for winning model” – I guess you mean the model that best fits the data rather than the one that results in optimal performance? (also either “a” or “the” is missing).

Figure 6: even for feeder absent trials the log-likelihood is mostly above zero indicating feeder present in my understanding. I think this is why the “decision threshold” needs to be optimized to match the data, but this seems like a big problem with this and many of the other models that should be stated clearly (if you expected this then say why) and discussed in the discussion. In particular in Supp Fig 18 it seems that for whole families of models the decision criterion is either always positive or always negative – to help the reader these qualitative changes across models should be explained. i.e. there must be a reason why some models are biased toward positive and others to negative likelihood ratios. The explanation on p.16 (e.g. different motor costs) is unsatisfactory as that would be revealed in a behavioral bias rather than a model bias and in any case does not say why different families of models would err in different directions (or why does the Bayesian model need a specific N-dependence, from the point of view of what is going on within the model, not just the observed output). That is to say, irrespective of behavior, what is going on in the models to require such biases? Or are different “d” values only needed in the model to explain behavioral biases and all models produce greater accuracy with a d of zero? A lot more explanation is needed concerning this issue.

Discussion: I would like to see a bit more focus on what is new/surprising, why some calculations are feasible in the brain, not others, what calculations (e.g. agglomerative clustering) have any evidence for them in other literature and so on. Are there ways of producing stimuli more artificially that would hinder one of the strong contender models more than another, and so on. As of now it does not seem like there is a strong takeaway. My understanding is that the full Bayesian method would be intractable anyway so I am not sure it was ever a serious contended, but here it is good to see some evidence disfavoring it. But again, some more discussion about the “false priors” results is needed. Do these really rescue the Bayesian model? Is there any reason to think that the subject would know the exact priors so not produce false priors? If not, perhaps the “false priors” are optimal given the data in some sense, so would that not make them a “more Bayesian” version than the standard?

As you can see, with the wealth, even overabundance, of tests and results, a lot of questions are raised and the paper would be more satisfactory if a sizable subset of potential questions are discussed and addressed in the text.

Finally, looking at the GitHub repository with the code, the ReadMe is insufficient to help anyone run the code (there are so many files I suspect it would be practically impossible for anyone to run it, so without instructions it is in practice unavailable).

Minor issues:

p. 5 “Family B” is used when I think you are still describing Family C (right after the eq.).

top of p.12 a weird symbol instead of “<” for the p-value

Eq.23: I think N_0 and N_1 should be defined though one can guess they are numbers non/affiliated with the feeder in a given trial.

Before eq.24 a missing close parenthesis

p.20 “see … box” it would be nice to tell the reader where to find the box as it is not nearby.

**Have the authors made all data and (if applicable) computational code underlying the findings in their manuscript fully available?**

Reviewer #1: Yes

Reviewer #2: **No: **See above -- present but impractical to use without instructions.

PLOS authors have the option to publish the peer review history of their article (what does this mean?). If published, this will include your full peer review and any attached files.

Reviewer #1: No

Reviewer #2: No

Figure Files:

Data Requirements:

Reproducibility:

References:

---

## [Editor Report · Decision Letter 1]

5 Oct 2021

Dear Ms Lee,

We are pleased to inform you that your manuscript 'Point-estimating observer models for latent cause detection' has been provisionally accepted for publication in PLOS Computational Biology.

Best regards,

Jean Daunizeau

Associate Editor

PLOS Computational Biology

Samuel Gershman

Deputy Editor

PLOS Computational Biology

---

## [Editor Report · Acceptance letter]

25 Oct 2021

PCOMPBIOL-D-21-01027R1 

Point-estimating observer models for latent cause detection

Dear Dr Lee,

I am pleased to inform you that your manuscript has been formally accepted for publication in PLOS Computational Biology. Your manuscript is now with our production department and you will be notified of the publication date in due course.

With kind regards,

Andrea Szabo
